# Dopamine and Beyond: Implications of Psychophysical Studies of Intracranial Self-Stimulation for the Treatment of Depression

**DOI:** 10.3390/brainsci12081052

**Published:** 2022-08-08

**Authors:** Vasilios Pallikaras, Peter Shizgal

**Affiliations:** Centre for Studies in Behavioural Neurobiology, Department of Psychology, Concordia University, Montreal, QC H4B 1R6, Canada

**Keywords:** brain-stimulation reward, medial forebrain bundle, convergence model, treatment-resistant depression

## Abstract

Major depressive disorder is a leading cause of disability and suicide worldwide. Consecutive rounds of conventional interventions are ineffective in a significant sub-group of patients whose disorder is classified as treatment-resistant depression. Significant progress in managing this severe form of depression has been achieved through the use of deep brain stimulation of the medial forebrain bundle (MFB). The beneficial effect of such stimulation appears strong, safe, and enduring. The proposed neural substrate for this promising clinical finding includes midbrain dopamine neurons and a subset of their cortical afferents. Here, we aim to broaden the discussion of the candidate circuitry by exploring potential implications of a new “convergence” model of brain reward circuitry in rodents. We chart the evolution of the new model from its predecessors, which held that midbrain dopamine neurons constituted an obligatory stage of the final common path for reward seeking. In contrast, the new model includes a directly activated, non-dopaminergic pathway whose output ultimately converges with that of the dopaminergic neurons. On the basis of the new model and the relative ineffectiveness of dopamine agonists in the treatment of depression, we ask whether non-dopaminergic circuitry may contribute to the clinical efficacy of deep brain stimulation of the MFB.

## 1. Introduction

Major depressive disorder is a heterogeneous chronic mental disorder that leads to high rates of functional impairment and is involved in at least half of completed suicides [1]. Epidemiological data since the 1950s report that the prevalence of depression has remained stable or has increased [2,3,4]. Moreover, over the last two decades, the rates of both moderate and severe depression have doubled in US adolescents as depression is diagnosed earlier in life [5]. During the COVID-19 pandemic, the prevalence of the disorder rose steeply; a systematic review across 204 countries described a 27.6% increase in the global prevalence of depression compared to prior estimates [6]. These concerning epidemiological patterns continue to unfold despite the discovery and use of several effective interventions, ranging from the first-generation antidepressants in the 1950s to new applications of psychosurgery and psychedelic drugs in recent decades [7]. As depression continues to afflict a large proportion of individuals worldwide, a lot remains to be learned about its pathophysiology and treatment [8].

An estimated 3 out of 10 individuals diagnosed with depression suffer from symptoms that do not improve after consecutive rounds of treatment [9]. These patients are classified as having treatment-resistant depression. The discovery that deep brain stimulation is an effective intervention in the most severely afflicted members of this population [10] provides grounds for optimism about further development of successful treatments. Amongst the several brain areas that have been investigated as deep brain stimulation targets for depression, the medial forebrain bundle (MFB) shows particular promise [11,12,13]. Across multiple studies, patients with treatment-resistant depression receiving continuous bilateral MFB deep brain stimulation experience immediate and long-term symptom improvement with a high response rate [12].

Given the estimated 280 million adults who suffer from depression [14] and the expertise and cost involved in deep brain stimulation surgery and follow-up care, large-scale application of this intervention is untenable. That said, the impressive clinical efficacy of MFB deep brain stimulation provides a guidepost to the critical neural circuitry underlying depression, long sought but yet to be identified definitively. Understanding how MFB deep brain stimulation achieves antidepressant efficacy may lead to studying other, less invasive, neuromodulation methods amenable to larger-scale applications. A first step in this direction is to delineate the neuroanatomy of the MFB fibers involved in the antidepressant effect. Although a lot of attention has been paid to midbrain dopamine neurons, researchers have reported that MFB deep brain stimulation leads to activation of both dopaminergic and non-dopaminergic pathways [12,15].

In this white paper, we expand on our recent theory article [16] to explore the idea that both dopaminergic and non-dopaminergic neural pathways may underlie the antidepressant effect of MFB deep brain stimulation. We discuss animal models of depression, the effect of dopamine agonists in depression, the biophysical properties of different axons traversing the MFB, and findings on the brain reward system architecture supporting the parallel contributions of converging dopaminergic and non-dopaminergic inputs to the final common path for reward seeking.

## 2. Animal Models of Depression: The Bright Side

Behavioral neuroscientists use animal models to study psychopathology, including depression. Proponents of animal models argue that they facilitate the study of brain–behavior relations under conditions of experimental control over extended periods of time [17,18]. Critics have raised concerns about the reliance on arguments from analogy to extrapolate to human psychopathology [19,20]. This view suggests that neuroscientists may often commit “the modeler’s functional fallacy” by overlooking fundamental differences between the models and the target (i.e., psychopathology) pertinent to transfer of results to humans [21]. In other words, the changes produced via experimental control of environmental or genetic factors in laboratory animals do not necessarily map onto human etiology or symptomatology because these designs cannot mimic the complexity of human conditions [20]. For example, a model that is widely used to study depression in animals is exposure to stress. Indeed, experiencing serious stressors poses a significant risk for developing mental illness [22]. However, life stressors appear to contribute to a non-specific predisposition to the development of psychopathology and bodily illnesses alike [23]. Further, a large proportion of individuals who undergo traumatic experiences do not develop any psychopathology [24]. Thus, the reasoning that stress can be used to model depression in animals may be unsound.

Notwithstanding the difficulties in establishing direct causal relationships, animal models are useful in generating hypotheses that should be tested and scrutinized in humans [20]. In the absence of simple solutions, a pathway to generate hypotheses is to focus on constructs that can be studied in both humans and animals [25]. Still, the identification, operationalization, and measurement of such constructs is not straightforward [19]. In the case of depression, motivational deficits have been described across the entire developmental trajectory of depression, ranging from at-risk offspring, to patients with active depression and patients with residual symptoms [26,27,28]. As a result, we believe that basic scientific findings on the structure and function of the brain reward system may provide a wealth of etiological and treatment hypotheses that can be tested in depressed individuals. In this way, neuroscientists can capitalize on a host of advanced neuroanatomical and behavioral methods that cannot be used in humans to study motivation, an environmentally preserved process across species.

A powerful means for studying motivation is provided by the intracranial self-stimulation (ICSS) phenomenon. Interestingly ICSS has been recognized since the 1980s as a valid model for studying depression in animals [29]. However, in the context of depression, much less attention has been paid to this methodology compared to the stress-induction models. We suggest that combining ICSS and novel neuroanatomical methods to uncover the function and architecture of the brain reward system may be a prerequisite to understanding the motivational deficits in depression. We now turn to the role of dopamine systems in depression, which serves as a foundation for the link between research on brain stimulation reward and the clinical effect of electrical MFB stimulation.

## 3. Dopamine Transmission and Depression

The study of the neurochemical correlates of depression accelerated in the 1950s with the serendipitous discoveries that a variant of an anti-tuberculosis agent (iproniazid, a monoamine oxidase inhibitor) and a drug developed as an antihistamine (imipramine, a tricyclic antidepressant) have antidepressant properties [7,30]. Based on the pharmacodynamics of these drugs, researchers formulated the monoamine hypothesis of affective disorders, according to which depression is associated with a synaptic deficit of one or more monoamines [31,32]. This influential theory sparked a literature evaluating the role of synaptic monoamine availability in depression and led to the development of selective serotonin reuptake inhibitors, the current first-line antidepressants that have high clinical efficacy and fewer side effects than earlier drugs [30].

Researchers have paid significant attention to the role of another monoamine, dopamine, in depression. An extensive literature suggests that the dopamine system may be implicated in the pathophysiology of depression, and the mechanism subserving several interventions, including deep brain stimulation [15,33,34]. The hypothesis that agents boosting dopamine transmission (e.g., psychostimulants) can improve mood has been researched for close to a century [35]. As we review in our recent article [16], although psychostimulants appear to induce a rapid mood improvement, this effect is transient. Clinical recommendations for depression suggest that stimulants should be used seldomly as adjuncts to conventional pharmacotherapy with the goals of addressing fatigue and alertness, two well-established clinical endpoints of psychostimulants [36]. Additionally, psychostimulants are the first-line pharmacotherapy for attention deficit hyperactivity disorder (ADHD), a condition that has high comorbidity rates with depression [37]. Adding to the notion that dopamine agonists may be ineffective at managing depression, conventional antidepressants, and not psychomotor stimulants, are recommended for the treatment of moderate to severe comorbid depression to ADHD [38].

The rapid mood improvement induced by psychostimulants is a main reason for the continued interest in their role in treating depression [36]. Thus, a sustained increase in midbrain dopamine system activity may be required to induce an enduring mood improvement. This hypothesis is congruent with the sustained increase in midbrain dopamine activity caused by standard deep brain stimulation protocols, which entail continuous stimulation [12]. However, a randomized controlled trial investigating the antidepressant efficacy of an extended-release form of methylphenidate as adjunct treatment for depression failed to detect a clinical effect [39]. Although further research is needed to determine whether extended-release stimulants and/or dosing may produce a sustained clinical effect, the current consensus holds that psychostimulants are insufficient as a monotherapy for depression. This brings into question the role of midbrain dopamine firing in the strong clinical effect of MFB deep brain stimulation. We address this question through the lens of the convergence model of brain reward circuitry [40], thus asking whether midbrain dopamine activation by MFB deep brain stimulation is necessary and/or sufficient for the antidepressant effect.

## 4. Brain Stimulation Reward

The first report of ICSS was roughly contemporaneous to the first pharmacological findings concerning antidepressant action. Olds and Milner showed that animals will work tirelessly to trigger electrical stimulation of various deep brain loci [41]. Their discovery launched the study of brain reward circuitry. With the rewarding effect of MFB stimulation in rodents as our focus, we now discuss how findings from the basic and clinical literature may converge to guide the identification of the neurons subserving the rewarding and antidepressive effects.

A rat working for rewarding electrical stimulation of the MFB provides a mirror image of depression. Whereas a depressed patient will struggle to summon the motivation to pursue previously rewarding activities (“motivational anhedonia” [42,43]), the behavior of the rat is strongly energized by the recent receipt of highly rewarding stimulation (the “priming effect” [44,45,46,47]). Whereas a depressed patient will manifest pessimistic expectations about future rewards [48], the rat shows enthusiastic anticipation of imminent stimulation trains, which it will pursue in preference to biologically essential resources such as food and water [47,49].

Papers co-authored by Jaak Panksepp and pioneers in the use of deep brain stimulation to relieve treatment-resistant depression propose a fundamental link between the neural mechanisms underlying the clinically beneficial effect in humans and intracranial self-stimulation in rodents [13,50,51]. Midbrain dopamine neurons figure prominently in accounts of both phenomena. Here, we briefly summarize evolving views of the role of dopamine in intracranial self-stimulation, and we discuss the potential implications of recent findings in rodents to understand how MFB stimulation provides relief of treatment-resistant depression.

We discern three phases in how the role of dopaminergic transmission in brain stimulation reward has been understood. In the first two phases, a small core of experts assessed available data with caution and nuance that tended to diminish as the distance between an observer and the primary data increased. We hope that this commentary can contribute to forestalling a similar trend should the recently emerged third phase attract the attention of our colleagues.

### 4.1. Phase 1: Direct Stimulation of Dopaminergic Axons

The first phase followed in the wake of a momentous neuroanatomical advance: the use of histochemical fluorescence to provide the first visualization of central neural projections on the basis of their neurochemistry [52,53]. The resulting depictions of monoaminergic pathways (e.g., Figure 1) galvanized neuroscientific research at the levels of molecules, cells, circuits, and behavior. The clustering of positive self-stimulation sites along the trajectories of the ascending catecholaminergic pathways was noted repeatedly [54,55,56], and dopamine came to gain the upper hand over noradrenaline [57,58]. Although both the correlational nature of the anatomical evidence and exceptions to the pattern were noted by the experts, a cruder conclusion gained widespread traction and has continued to be expressed in some textbooks, print media, and internet sources. On that view, electrical stimulation of sites along the MFB is rewarding because of the direct excitation of the dopaminergic axons.

### 4.2. Phase 2: The Series-Circuit Model

The second phase of conceptualization of the role of dopaminergic axons in MFB self-stimulation incorporates the results of psychophysical studies that estimated the refractory periods, conduction velocity, and behaviorally relevant direction of conduction of the fibers directly responsible for the rewarding effect [60,61,62,63,64,65]. Those studies implicate neurons with myelinated axons and a rostro-caudal direction for the reward-related signals. These properties contrast sharply with those of the rodent midbrain dopamine neurons, which give rise to fine, ascending, unmyelinated axons that conduct slowly and recover from refractoriness too late to account for the psychophysically derived estimates [66,67,68,69,70,71,72].

The psychophysical characterization of the directly stimulated neurons subserving MFB self-stimulation opposes the hypothesis that direct activation of the dopaminergic axons is the sole (or even principal) cause of the rewarding effect. If so, how can the extensive pharmacological data implicating dopaminergic neurotransmission in ICSS be accommodated? The “series-circuit” model (Figure 2) was proposed to accomplish this [63,73]. In this view, myelinated, relatively fast-conducting fibers constitute all or most of the directly stimulated substrate, which provides synaptic input to the midbrain dopamine cell bodies; trans-synaptic excitation of VTA dopamine cell bodies and the ensuing efferent consequences give rise to a rewarding effect.

According to the series-circuit model, the directly stimulated neurons subserving MFB self-stimulation should be sought among the mono- or poly-synaptic [65,74,75] inputs to dopamine somata in the VTA and substantia nigra pars compacta (SNc). The development of deletion-mutant rabies viruses incorporating eYFP has made it possible to visualize the mono-synaptic inputs to these cell bodies (Figure 3). This depiction, along with others [76] that include the remarkable divergence of the dopaminergic projections (particularly those of the SNc neurons [77]), emphasize the status of the midbrain dopamine neurons as hubs whose inputs allow them to collect information from diverse sources and to broadcast their outputs widely. In this view, determining the function of any particular input or subset thereof would seem to be of secondary importance to the core question: the function(s) of the midbrain dopamine neurons themselves.

The importance of identifying the directly stimulated neurons subserving electrical self-stimulation of the MFB was further eclipsed by the demonstration that rodents will work vigorously for direct, specific, optogenetic activation of midbrain dopamine neurons. Even to experts, there are striking similarities between the behavior of rodents working for electrical MFB stimulation and optical stimulation of midbrain dopamine neurons, an observation that fits the series-circuit model neatly. In this view, the rewarding optical stimulation achieves directly what the electrical stimulation achieves indirectly: excitation of the midbrain dopamine neurons.

### 4.3. Phase 3: The Convergence Model

Despite the intuitive appeal of the series-circuit model and its acceptance beyond the circle of initial proponents, discrepant findings have long been noted. Huston and colleagues demonstrated that rats can learn to perform simple movements to earn MFB stimulation, despite extensive damage to the forebrain terminal fields of the midbrain dopamine neurons [79,80]. The rewarding effectiveness of such stimulation was little affected by large, bilateral, excitotoxic lesions of the nucleus accumbens terminal field [81]. Discrepancies were reported between the frequency dependence of the rewarding effect and of concomitant dopamine release [82].

The evidence most devastating to the series-circuit model arises from the recent measurement of operant performance as a function of both the strength and cost of reward [40]. The resulting three-dimensional depiction of the behavioral data resembles the corner of a plateau and has been dubbed the “reward mountain.” Displacement of this structure along the strength and cost axes are interpreted by means of a model built around a set of psychophysical functions [40,83,84,85]. These map the stimulation parameters, task requirements, and test-cage affordances into corresponding subjective variables. Of paramount importance among these is the “reward-growth” function that translates the aggregate firing rate [62,86] and duration [87] of a pulse train into the intensity of the rewarding effect. With the train duration held constant, this function is roughly sigmoidal. Displacement of the mountain along the stimulation-strength axis by a manipulation such as drug administration reflects an action prior to the input to the reward-growth function. If the function is shifted leftwards by a drug, this means that the stimulation now produces “more bang for the buck.” Such a shift would be reflected in a reduction in the pulse frequency, current, pulse duration, or train duration required to produce a given level of reward intensity. In contrast, drug action at or beyond the output of the reward-growth function shifts the reward mountain along the cost axis. The stimulation strength required to produce a reward of half-maximal intensity is unaltered by such an effect, but the rewarding impact of all pulse frequencies has been scaled upwards or downwards. To hold performance constant, a compensatory adjustment of reward cost is required.

Manipulations of dopaminergic neurotransmission have consistently displaced the reward mountain along the cost axis and have only rarely been accompanied by shifts along the stimulation-strength axis [85,88,89,90]. Accommodating these results within the series-circuit model requires that a reward-growth function be positioned between the output of the directly stimulated neurons and the ***input*** to the dopamine neurons.

The key evidence that falsifies the series-circuit model was provided by a reward-mountain study in which direct, specific, optogenetic activation of midbrain dopamine neurons was substituted for electrical MFB stimulation [40]. Administration of a specific dopamine-transporter blocker shifted the reward mountain along ***both*** the stimulation strength and cost axes. Shifts along the strength axis require that a non-linear reward-growth function be positioned at or beyond the ***output*** of the dopamine neurons. However, positioning a non-linear reward-growth function downstream from the dopamine neurons falsifies the predictions of the series-circuit model in the case of electrical MFB stimulation. A non-linear growth function at or beyond the output of the dopamine neurons predicts shifts of the reward mountain along the stimulation-strength axis in response to drugs that perturb dopaminergic neurotransmission. This is not what was seen in 27 of 32 rats tested in 4 studies [85,88,89,90]. In short, the series-circuit model cannot handle the differences observed in the response of the reward mountain to dopaminergic challenge in rats working for electrical MFB stimulation and specific, direct, optical activation of midbrain dopamine neurons.

A new model of the brain circuitry subserving the reward-seeking behavior was proposed to reconcile the data from the electrical and optical self-stimulation studies [40]. In this model (Figure 4), at least two separate neural populations subserve the rewarding effects, and their outputs ultimately converge on the behavioral final-common path for reward evaluation and pursuit. Neurons that give rise to myelinated MFB fibers carry the principal component of the reward signal directly induced by electrical MFB stimulation despite the fact that the stimulation does produce some trans-synaptic activation of midbrain dopamine neurons. At stimulation strengths that produce a robust reward in response to electrical MFB stimulation, the model holds that this trans-synaptic dopaminergic activation is generally too weak to exert a detectable influence on the position of the reward mountain.

Figure 4 provides only a simplified depiction of the convergence model. In the full model [16,40], there is additional interaction between dopaminergic neurotransmission and the reward signal induced by electrical MFB stimulation. The parallel signal flow in the myelinated MFB axons and the midbrain dopamine neurons coupled with the later interaction between these two channels raise new possibilities for interpreting the therapeutic effect of MFB stimulation in humans.

Following the receipt of highly rewarding electrical MFB stimulation, rats show powerfully enhanced motivation to obtain additional stimulation trains [44,45,47]. This potentiating “priming” effect appears to survive blockade of D2 dopamine receptors [49]. Thus, this effect may well arise from activation of the upper (myelinated axon) limb of the model in Figure 4. New developments in tract-tracing methods coupled with the power of optogenetic methods to test the sufficiency and necessity of identified neural pathways promise to provide the means for identifying the neurons that give rise to the myelinated axons in question.

## 5. Contribution of Midbrain Dopamine Neurons to the Treatment of Depression: One Pathway among Many?

At the phenomenological level, the priming effect would appear well suited to counteract the motivation anhedonia typical of depression. If so, finding the directly activated neurons subserving this effect in rodents would raise important questions for research aimed at determining the neural mechanisms responsible for the beneficial effect of MFB stimulation in relieving treatment-resistant depression. Does a homologous pathway exist in humans and pass sufficiently close to the clinically effective stimulation site to be activated by the antidepressant stimulation? If these questions are answered in the affirmative and the neurotransmitter(s) released by the neurons constituting this pathway are identified in rodents, this could point the way towards a new pharmacological tool for treating depression.

The convergence model brings to mind images of MFB circuitry (Figure 5) much more extensive than those in Figure 1 and Figure 3. Figure 5 captures the remarkable number of origins and termination fields of MFB axons in the rodent. The dopaminergic component, so memorable in Figure 1, is sometimes portrayed as isomorphic to the MFB. Figure 5 shows that the dopaminergic projections constitute merely one subset among many. We ask whether this subset is truly “primus inter pares” or whether additional components that have yet to be fully characterized may rival its functional importance.

The notion that signals from the myelinated axons subserving MFB self-stimulation reach the behavioral final-common path for reward pursuit in parallel to those from dopamine neurons (Figure 4) invites us to consider MFB outputs that extend beyond the midbrain region housing dopaminergic somata. Of particular interest are the branching projections described by Nauta and Domesick [92]. They note a large contingent of MFB axons that execute a turn in the VTA and pass caudo-laterally just above the SNc; a substantial subset of these fibers then execute another turn and head dorsomedially. Some terminate in the midbrain reticular formation, whereas others continue to course caudally into the ventral portion of the central grey. Fibers in a second more medial division pass through the VTA, and some reach the laterodorsal tegmental nucleus, ventral portion of the central grey, and locus coeruleus. Could axons in one or both branches contribute to the therapeutically beneficial effect of MFB stimulation in humans, in parallel to, or even instead of, the contribution of the MFB inputs to the midbrain dopamine neurons? Along the trajectories of the branches described by Nauta and Domesick, we have recorded compound action potentials elicited by rewarding stimulation at the lateral hypothalamic level of the MFB; recovery from refractoriness in the fibers from which the recordings were obtained corresponds to psychophysical measurements derived from the behavior of the rats [93].

On the basis of data acquired in human patients by means of diffusion-weighted MRI and supplemented by neuroanatomical staining of post-mortem tissue, Coenen and colleagues have proposed that a principal substrate for the antidepressant effect of MFB stimulation in humans is composed of myelinated fibers arising in the orbitofrontal cortex and terminating on dopaminergic somata in the VTA [94]. We understand that diffusion tractography is crucial to the success of the surgical approach and that there is often correspondence between fiber trajectories inferred by this method and those measured directly by conventional neuroanatomical means. Nonetheless, the interpretation of diffusion-weighted MRI data is dependent on assumptions that do not always hold [95]. This is particularly so when different fiber tracts cross, branch, abut, and/or bend [95,96,97], as is the case of the MFB circuitry described by Nauta and Domesick.

Based on the third-phase view of the role of dopaminergic neurons in MFB self-stimulation, we appeal to our colleagues to entertain the possibility that a reward-related pathway that parallels the midbrain dopaminergic projections is activated by the clinically effective stimulation and contributes to the clinically beneficial effect.

We appeal further for additional neuroanatomical studies of the region of the human MFB where electrical brain stimulation relieves depressive symptoms. Although we know of no studies demonstrating myelination of dopaminergic axons in rodents, some axons arising in the SNc of the squirrel monkey are myelinated [98]. Do any VTA dopamine neurons in the human give rise to myelinated axons? If so, where do their calibers fit within the spectrum of myelinated fibers at the critical brain site?

## 6. Conclusions

In addition to its implications for identifying the cellular substrates of the antidepressant effect of MFB stimulation, the third-phase view also sends a more general message, one concerning convergent causation (“equifinality”). The motor patterns that constitute goal-seeking actions are highly stereotyped and are subject to multiple influences. The fact that an animal performs such actions does not tell us which subset of these multiple influences is responsible. Although there is a striking behavioral resemblance between rats working for electrical MFB stimulation and optical stimulation of midbrain dopamine neurons, the research that gave rise to the convergence model implies that partially different causal chains link the stimulation to the behaviors. In this view, both dopaminergic and non-dopaminergic pathways, operating partially in parallel, subserve ICSS of the MFB. Given the important role of appetitive motivation in ICSS and depression, both pathways should be considered as candidate neural substrates for the relief of treatment-resistant depression by electrical MFB stimulation.

## Figures and Tables

**Figure 1 brainsci-12-01052-f001:**
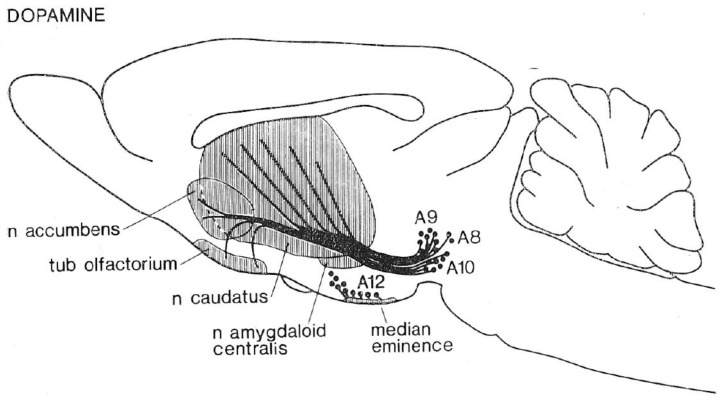
Ascending projections of midbrain dopamine neurons (from [59]).

**Figure 2 brainsci-12-01052-f002:**
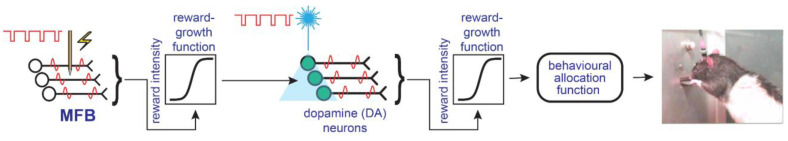
The series-circuit model (from [16]).

**Figure 3 brainsci-12-01052-f003:**
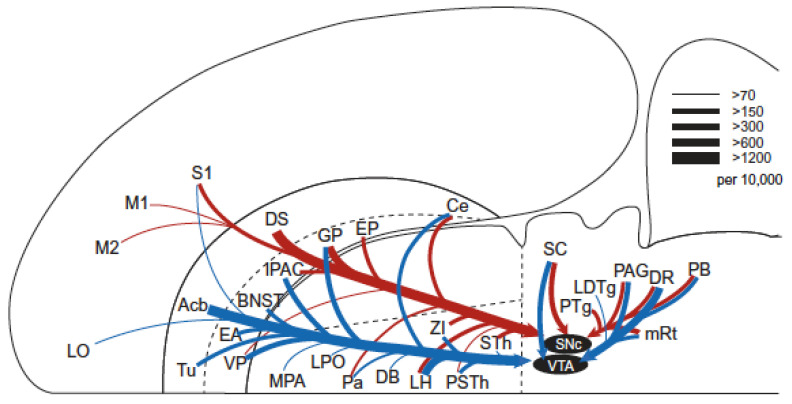
Horizontal-plane depiction of mono-synaptic inputs to VTA and SNc dopamine cell bodies (from [78]).

**Figure 4 brainsci-12-01052-f004:**
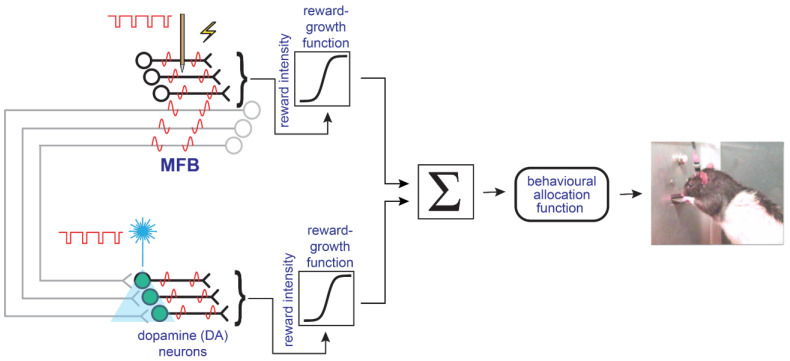
The convergence model simplified. Myelinated, non-dopaminergic axons and midbrain dopamine neurons provide converging inputs to the final-common path for reward evaluation and pursuit. From [16].

**Figure 5 brainsci-12-01052-f005:**
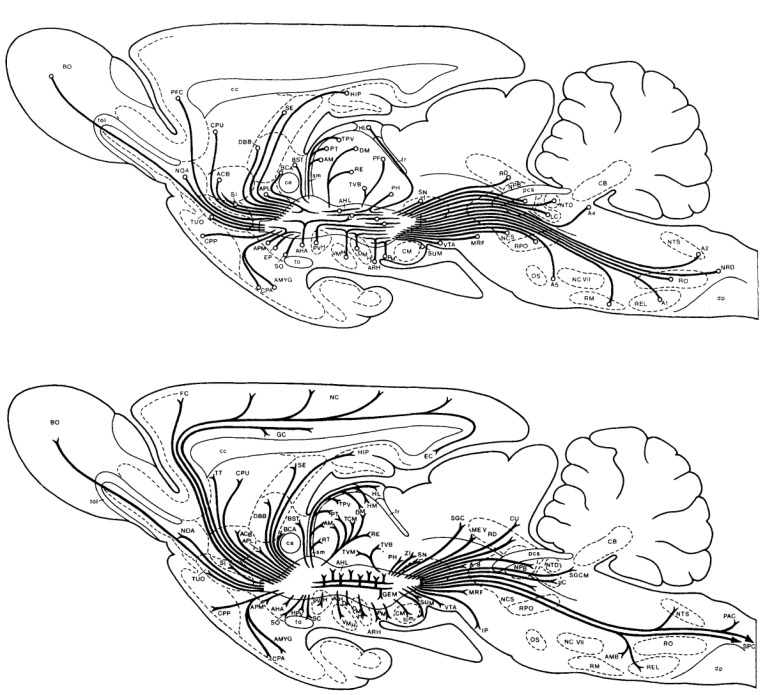
Inputs to (**top**) and outputs from (**bottom**) the rat MFB, from [91].

## Data Availability

Not applicable.

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
