# Peer review of "Dopamine and Beyond: Implications of Psychophysical Studies of Intracranial Self-Stimulation for the Treatment of Depression"

_brainsci, 2022, doi:10.3390/brainsci12081052_

Round 1
Reviewer 1 Report
In this article, the authors discuss three different models regarding the role of dopamine neurons in intracranial self stimulation (ICSS). The authors use ICSS to explain motivational deficits in depression. However, ICSS has been studied in the context of addiction. Animals would forgo food or water for self stimulation. Such intense appetitive responses do not explain the motivational deficits observed in depression. Animal models for testing antidepressants use motivation/helplessness models such as forced swim test or tail suspension test. Thus the relation between ICSS and depression seems weak.
However, within the context of ICSS, the models are explained well and will be useful to the field.
Minor correction: Line 219: encorporating -> incorporating
Author Response
Thank you for your helpful review of our paper and for your assessment that the models we discuss have been explained well.
As you point out, we relate motivational deficits in depression to ICSS. The key link is our view that the self-stimulating rat “provides a mirror image of depression.” We observe:
"Whereas a depressed patient will struggle to summon the motivation to pursue previously rewarding activities (“motivational anhedonia” [42,43]), the behavior of the rat is strongly energized by the recent receipt of highly rewarding stimulation (the “priming effect” [44–47]). Whereas a depressed patient will manifest pessimistic expectations about future rewards [48], the rat shows enthusiastic anticipation of imminent stimulation trains which it will pursue in preference to biologically essential resources such as food and water [47,49]. "
* Reference numbers refer to the revised manuscript
You point out that: “Animal models for testing antidepressants use motivation/helplessness models such as forced swim test or tail suspension test.” Indeed, much work has been carried out in that vein. Our observation pertains to a complementary perspective pioneered by Jaak Panksepp and expressed in the early papers of Coenen and co-workers on the relief of treatment-resistant depression by deep-brain stimulation of the MFB. In that work (see added reference 13), Panksepp’s SEEKING system links ICSS to the antidepressant action of deep-brain stimulation of the MFB as well as to our “mirror-image” analogy. Thus, we believe that there are firm grounds for relating ICSS to depression.
In the section called “Animal Models of Depression: the Bright Side,” we highlight how a dimensional perspective of psychopathology can inform animal models of depression. We hope that the addition of reference 13 will help clarify and support our view that the ICSS phenomenon in laboratory animals and depression in humans are linked via common motivational processes.
Thank you for pointing out the typographical error, which we have now corrected.
Reviewer 2 Report
The manuscript entitled “Dopamine and Beyond: Implications of Psychophysical Studies of Intracranial Self-Stimulation for the Treatment of Depression” by Pallikaras and Shizgal is focusing an interesting approach and actual topic since MDD still has a high incidence worldwide.
The manuscript in the form of Perspective is well-written and organized. The authors highlighted the relevant information and covered the subject with adequate literature data.
The authors mentioned that permission for some of the Figures has not been requested yet. However, they should provide permissions or replace them.
Conclusion section resembles some kind of questionnaire. Although it is competent, the authors should rephrase those sentences and allow the positive statements.
Check the text for typing errors.
Author Response
Thank you for your helpful comments.
The permission issues have now been resolved. We have secured the rights to reproduce Figures 1, 3, & 5 by means of the RightsLink mechanism specified by the publishers (Wiley and Cell Press). Figures 2 and 4 were published in open-source journals that allow authors to reuse figures in other publications providing that they are referenced accordingly.
As you have suggested, we have rephrased the conclusions, replacing the interrogative form of several sentences by the declarative form. The conclusions now read as follows:
“In addition to its implications for identifying the cellular substrates of the antidepressant effect of MFB stimulation, the third-phase view also sends a more general message, one concerning convergent causation (“equifinality”). The motor patterns that constitute goal-seeking actions are highly stereotyped and are subject to multiple influences. The fact that an animal performs such actions does not tell us which subset of these multiple influences is responsible. Although there is a striking behavioral resemblance between rats working for electrical MFB stimulation and optical stimulation of midbrain dopamine neurons, the research that gave rise to the convergence model implies that partially different causal chains link the stimulation to the behaviors. On that view, both dopaminergic and non-dopaminergic pathways, operating partially in parallel, subserve ICSS of the MFB. Given the important role of appetitive motivation in ICSS and depression, both pathways should be considered as candidate neural substrates for the relief of treatment-resistant depression by electrical MFB stimulation.”
Reviewer 3 Report
The authors evaluated an interesting topic that is worth careful analysis. The authors offered an explanation of how medial forebrain bundle stimulation provides relief for treatment-resistant depression and the mechanisms underlying the antidepressant effect of deep-brain stimulation.
However, I am confused with the concept of presented figures since that they are either from their previous papers or they are without permission. Anyway, the authors should solve that issue prior to submission.
I really endorsed the competence of the authors.
Still, I strongly suggest that their dilemmas in the Conclusion section should be completely rephrased.
Author Response
Thank you for your helpful comments and endorsement.
As you have suggested, we have rephrased the conclusions, replacing the interrogative form of several sentences by the declarative form. The conclusions now read as follows:
“In addition to its implications for identifying the cellular substrates of the antidepressant effect of MFB stimulation, the third-phase view also sends a more general message, one concerning convergent causation (“equifinality”). The motor patterns that constitute goal-seeking actions are highly stereotyped and are subject to multiple influences. The fact that an animal performs such actions does not tell us which subset of these multiple influences is responsible. Although there is a striking behavioral resemblance between rats working for electrical MFB stimulation and optical stimulation of midbrain dopamine neurons, the research that gave rise to the convergence model implies that partially different causal chains link the stimulation to the behaviors. On that view, both dopaminergic and non-dopaminergic pathways, operating partially in parallel, subserve ICSS of the MFB. Given the important role of appetitive motivation in ICSS and depression, both pathways should be considered as candidate neural substrates for the relief of treatment-resistant depression by electrical MFB stimulation.”